# Structure, Luminescence and Temperature Detection Capability of [C(NH_2_)_3_]M(HCOO)_3_ (M = Mg^2+^, Mn^2+^, Zn^2+^) Hybrid Organic–Inorganic Formate Perovskites Containing Cr^3+^ Ions

**DOI:** 10.3390/s23146259

**Published:** 2023-07-09

**Authors:** Dagmara Stefańska, Adam Kabański, Thi Hong Quan Vu, Marek Adaszyński, Maciej Ptak

**Affiliations:** Włodzimierz Trzebiatowski Institute of Low Temperature and Structure Research, Polish Academy of Sciences, 50-422 Wroclaw, Poland; a.kabanski@intibs.pl (A.K.); q.vu@intibs.pl (T.H.Q.V.); m.adaszynski@intibs.pl (M.A.); m.ptak@intibs.pl (M.P.)

**Keywords:** MOF, hybrid perovskite, luminescence, thermometry, chromium(III) ions, temperature sensing

## Abstract

Metal-organic frameworks are of great interest to scientists from various fields. This group also includes organic–inorganic hybrids with a perovskite structure. Recently their structural, phonon, and luminescent properties have been paid much attention. However, a new way of characterization of these materials has become luminescence thermometry. Herein, we report the structure, luminescence, and temperature detection ability of formate organic–inorganic perovskite [C(NH_2_)_3_]M(HCOO)_3_ (Mg^2+^, Mn^2+^, Zn^2+^) doped with Cr^3+^ ions. Crystal field strength (Dq/B) and Racah parameters were determined based on diffuse reflectance spectra. It was shown that Cr^3+^ ions are positioned in the intermediate crystal field or close to it with a Dq/B range of 2.29–2.41. The co-existence of the spin-forbidden and spin-allowed transitions of Cr^3+^ ions enable the proposal of an approach for remote readout of the temperature. The relative sensitivity (S_r_) can be easily modified by sample composition and Cr^3+^ ions concentration. The luminescent thermometer based on the ^2^E/^4^T_2g_ transitions has the relative sensitivity S_r_ of 2.08%K^−1^ at 90 K for [C(NH_2_)_3_]Mg(HCOO)_3_: 1% Cr^3+^ and decrease to 1.20%K^−1^ at 100 K and 1.08%K^−1^ at 90 K for Mn^2+^ and Zn^2+^ analogs, respectively.

## 1. Introduction

The noticeable development of hybrid organic–inorganic perovskites (HOIPs) has been observed in recent years. The materials with the general formula ABX_3_, where A is an inorganic or organic cation (e.g., NH_4_^+^, (CH_3_)_2_NH^+^), B is a divalent metal ion (e.g., Pb^2+^, Zn^2+^), and X a monovalent anion (e.g., Cl^−^, HCOO^−^) have attracted increasing attention due to their extraordinary properties [1,2,3]. Hybrid materials, e.g., CH_3_NH_3_PbCl_3_, have been particularly implemented in state-of-the-art photovoltaic devices [4,5,6]. However, their potential usefulness is significantly greater due to their characteristics, including ferroelectricity [7,8], magnetic [9], optoelectronic [4], and luminescent properties [10,11,12,13]. The characteristics of investigated materials can be widely tuned by the replacement of A, B, and X linkers [10,14].

Among various materials, the perovskite-like metal-organic frameworks (MOFs) containing formate anions (HCOO^−^) exhibit unique features, such as ferroelectricity, multiferroicity, and luminescence [1,10,15]. Particularly, the group of Cr-based materials shows strong luminescence and weak concentration quenching [2,10,15]. Nevertheless, temperature-dependent luminescence is one of the most outstanding phenomena. The temperature change induces the change in energy level populations, which makes formate-based compounds containing Cr^3+^ ions sufficient materials for non-contact temperature sensing [10].

The optical properties of the transition metals (TM), including chromium trivalent ions, can be affected by the crystal field (CF) strength [13,14,16]. The change in the CF strength leads to the change in the dominant transition type [17]. The Cr^3+^ ions luminescence may contain two particular emission bands: narrow spin-forbidden ^2^E_g_ → ^4^A_2g_ (around 700 nm) and broad spin-allowed ^4^T_2g_ → ^4^A_2g_ (around 750 nm). In low temperatures, the narrow ^2^E_g_ → ^4^A_2g_ emission is dominant. The increase in temperature induces the thermal population of the ^4^T_2g_ level and, consequently, promotes the broad ^4^T_2g_ → ^4^A_2g_ emission. The narrow emission takes place in a strong CF environment. The spin-allowed emission, in turn, occurs in a weak crystal field strength. The coexistence of both types of emission indicates the intermediate CF strength. The progressive increase in temperature leads to luminescence quenching. The significant influence of temperature on spectroscopic characteristics of Cr^3+^-based materials has become the basis of the thermometric model development [10,13].

Luminescence temperature sensing has attracted increasing attention recently [10,18,19,20,21,22]. Non-contact thermometry has great potential for application in scientific, industrial, and biomedical areas [23,24]. Among various advantages, the high accuracy and single measurement speed are noteworthy. The possibility of the plunge measurements going beyond typical thermal imaging limitations makes this approach a promising tool for industrial process monitoring [10].

Temperature sensing is mainly based on the detectable change in the luminescent properties induced by the change in the temperature. A thermometric model can be developed by monitoring changes in lifetime, peak position, as well as the insensitivities of specific peaks [20,25]. The comparison of the intensities of two temperature-dependent emission bands allows us to determine a thermometric parameter called fluorescence intensity ratio (FIR or Δ). Such an approach is called the ratiometric method and has been the most frequently reported application recently [18]. The methods relying on FIR analysis provide high sensitivity and make it possible to implement the independent sensing ranges, which leaves room for model optimization [13,26].

The vast majority of reported thermometric compounds are based on inorganic host materials with rare-earth (RE) elements as dopants [11,25,27,28]. However, the materials containing transition metal ions exhibit promising thermometric characteristics comparable to the solutions based on RE ions [29,30]. The highly sensitive thermometric properties have been reported, inter alia, for the perovskite materials containing ethylammonium cation and Cr^3+^ ions [10]. The development of luminescent thermometers based solely on chromium trivalent ions is a noteworthy approach enabling to deviate from the RE-based materials. Another notable strategy for the development of the ratiometric thermometer, presented in this work, is not only considering the luminescence of Cr^3+^ ions but also using the luminescence of the amine group, such as guanidinium cation ([C(NH_2_)_3_]^+^ denoted as GA^+^). The multicomponent thermometric model may be a promising approach toward higher sensitivity.

Herein, we report the synthesis as well as the structural and spectroscopic properties of the first metal–organic framework luminescent thermometers based on both GA^+^ and Cr^3+^ ion luminescence. Investigated series of [GA]M_1−x_Cr_x_(HCOO)_3_, where M = Mg^2+^, Mn^2+^, Zn^2+^, and x = 0, 0.01, 0.03, 0.05, have been synthesized and investigated as promising thermometric materials. The selection of three distinct cations was motivated by the fact that Mn^2+^ ions are the only ones that are optically active, and Zn^2+^ and Mg^2+^ ions create structures with different properties compared to transition metal ions such as Mn^2+^. All series exhibit outstanding temperature-dependent emission, which has become the basis of the thermometric analysis. This work is an attempt to describe the effect of the material composition on the luminescent properties with particular emphasis on luminescent thermometry. The optimization of the sensing range estimation is particularly considered. 

## 2. Materials and Methods

The starting materials include formic acid HCOOH (POCH, ≥98%), ethanol C_2_H_5_OH (POCH, 96%), guanidine carbonate salt [GA] [C(NH_2_)_3_]_2_CO_3_, (Sigma Aldrich, 99%) (Sigma Aldrich, Saint Louis, MI, USA), zinc(II) chloride ZnCl_2_ (Sigma Aldrich, 99.999%) (Sigma Aldrich, Saint Louis, USA), manganese(II) perchlorate hydrate Mn(ClO_4_)_2_⋅6H_2_O (Sigma Aldrich, ≥99%) (Sigma Aldrich, Saint Louis, USA), magnesium(II) chloride anhydrous MgCl_2_ (Sigma Aldrich, 99.9%), and chromium(III) chloride CrCl_3_ (Sigma Aldrich, 99%). All precursors were commercially available and were used for the synthesis without any further purification. In this study, a series of [GA]M_1−x_Cr_x_(HCOO)_3_ where M = Mn, Mg, Zn, and x = 0, 1%, 3%, 5%, were obtained by using the low-diffusion synthesis method. To grow [GA]M_1−x_Cr_x_(HCOO)_3_ crystals, at first formic acid (8.7 mmol) and GA (4.2 mmol) was dissolved in distilled water (20 mL). This solution was added by an aqueous solution (10 mL) containing 1.0 mmol of Mn(ClO_4_)_2_⋅6H_2_O/ZnCl_2_/MgCl_2_ for the pure samples. The amount of Cr^3+^ ions was calculated based on the molarity of the M^2+^ ions (see Appendix A). The resulting mixed solution was kept undisturbed and allowed to evaporate slowly. After two weeks, the crystals were harvested, washed with ethanol, and dried in the air. The color of the crystals was light pink for Mn or white for Mg and Zn. It also varied from green to dark green depending on the concentration of Cr^3+^ ions.

The powder X-ray diffraction (XRD) patterns were obtained on an X’Pert Pro X-ray diffraction system (Malvern Panalytical, Malvern, UK) equipped with a PIXcel detector (Malvern Panalytical, Malvern, UK) and using CuKα radiation (λ = 1.54056 Å). The Raman spectra were measured using a Bruker FT 110/S (Billerica, MA, USA) spectrometer operating at 1064 nm (Nd:YAG). The spectra were collected in a spectral range of 75–3200 cm^−1^ and with a spectral resolution of 2 cm^−1^. The diffuse reflectance spectra were obtained using a Varian Cary 5E UV–VIS–NIR spectrometer (Varian, Palo Alto, CA, USA). The temperature-dependent emission spectra were obtained with a Hamamatsu PMA-12 (Hamamatsu Photonics, Iwata, Japan) photonic multichannel analyzer combined with a BT-CCD sensor. As an excitation source, a 405 nm laser diode was used. The temperature was controlled by a Linkam THMS600 stage (Linkam, Tadworth, UK).

## 3. Results and Discussion

### 3.1. Structural Properties

The phase purity of all samples was confirmed by the XRD patterns with a simulation of the single-crystal structural data of [GA]Mn_1−x_Cr_x_(HCOO)_3_ (Figure 1). The samples with Mn^2+^ and Zn^2+^ crystallized in the orthorhombic *Pnna* crystal structure [31], and the details of the crystal structure of analogs with Mg^2+^ remain unknown. In general, the formate in-connected MnO_6_ framework crates cavities occupied by GA^+^ cations (see Figure 2). The right-shifting of the diffraction lines was observed due to the partial replacement of Mn^2+^ (CR = 81 Å), Mg^2+^ (CR = 86 Å), and Zn^2+^ ions (CR = 88 Å) by Cr^3+^ ions (CR = 75.5 Å). The crystal radius (CR) was obtained from Shannon [32]. No additional phases were detected, which indicates that the Cr^3+^ ions were substituted by the cation M.

The Raman spectra of the [GA]M_1−x_Cr_x_(HCOO)_3_ series, where M = Mg^2+^, Mn^2+^, Zn^2+^, and x = 0, 0.01, 0.03, 0.05, are marked in Figure 3a as GAMg, GAMn, and GAZn, respectively. All spectra are very similar and are consistent with the reported orthorhombic *Pnna* symmetry of all crystals [31,33,34,35]. However, some differences can be seen in the band shifts and the number of components, which are due to the different sizes, masses, and electronegativity of the metal cations that build the crystals. All these parameters affect the sizes of unit cells, causing Raman bands for [GA]M(HCOO)_3_ (M = Mg^2+^, Mn^2+^, Zn^2+^) to be shifted relative to each other (Figure 3b,c and Appendix A). Regarding [GA]Zn(HCOO)_3_, the upshifts observed for [GA]Mg(HCOO)_3_ are most pronounced for lattice modes observed below 300 cm^−1^ since they are very sensitive to the long-range order in the crystal. In addition, strong shifts towards higher wavenumbers, up to 12.3 cm^−1^ for [GA]Mg(HCOO)_3_ and 10.3 cm^−1^ for [GA]Mg(HCOO)_3_, are also observed for NH stretching vibrations above 2850 cm^−1^ (Figure 3d), which further indicate the weakest hydrogen bonds in the [GA]Mg(HCOO)_3_ crystal and stronger for [GA]Zn(HCOO)_3_. The upshift of 7.1 cm^−1^ when Zn^2+^ ions are replaced by Mg^2+^ was evidenced by bands associated with vibrations of oxygen atoms directly coordinated by metal ions, i.e., ν_2_ + ν_5_ that have been assigned to symmetric C–O stretching vibrations coinciding with C–H in-plane bending modes, respectively (Figure 3b) [36]. A much weaker upshift is observed for the stretching C–N modes, reaching 3.1 cm^−1^ for [GA]Mg(HCOO)_3_ and 3.2 cm^−1^ for [GA]Mn(HCOO)_3_ related to [GA]Zn(HCOO)_3_ (Figure 3a). This finding indicates very similar confinement of GA^+^ cations and similar dynamics in the perovskite void for M = Mg^2+^ and Mn^2+^.

The introduction of Cr^3+^ ions into the crystal structure of [GA]M(HCOO)_3_ (M = Mg^2+^, Mn^2+^, Zn^2+^) at such low concentrations causes very subtle effects on the spectra, not exceeding 1 cm^−1^. This confirms that aliovalent doping up to 5 mol% does not cause significant structural changes in the orthorhombic *Pnna* structure.

### 3.2. Optical Properties and Temperature Detection

The diffuse reflectance spectra (DRS) of representative samples [GA]M_1−x_Cr_x_(HCOO)_3_, where M = Mg^2+^, Mn^2+^, Zn^2+^, and x = 0.05, are shown in Figure 4. The intensity of the DRS spectrum is influenced by many factors, such as the size and position of crystallites [10]. Therefore, the DRS is used only for characterizing the localization of the energy levels of Cr^3+^ ions in each compound and the effect of the concentration of Cr^3+^ ions on the spectrum’s shape. Two primary broad bands localized around 16,828 cm^−1^ (594 nm) and 22,522 cm^−1^ (444 nm) for Zn-samples, 17,130 cm^−1^ (583.8 nm) and 23,162 cm^−1^ (431.7 nm) for Mg-samples, 17,050 cm^−1^ (586.5 nm) and 23,162 cm^−1^ (431.7 nm) for Mn-samples can be distinguished in Figure 3. These two bands are assigned to the spin-allowed transitions ^4^A_2g_ → ^4^T_1g_ and ^4^A_2g_ → ^4^T_2g_ of Cr^3+^ ions. In addition, a very weak and sharp peak centered at approximately 14,550 cm^−1^ (687.3 nm) is associated with the spin-forbidden transition from the ^4^A_2g_ ground state to the ^2^E excited level. It was found that when the concentration of Cr^3+^ increases, the position of the ^4^A_2g_ → ^2^E lines slightly changes (Appendix A). However, for the Zn-compounds, the ^4^A_2g_ → ^2^E absorption peak is invisible (Appendix A).

Noticeably, in the spectrum of Mn-samples, very weak and sharp peaks appeared at 29,240 cm^−1^ (342 nm) and 24,570 cm^−1^ (407 nm), which are attributed to the absorption of Mn^2+^ ions from ^6^A_1_ ground state to ^4^E, ^4^T_2_, and ^4^A_1_, ^4^E excited levels, respectively. The intensity of these bands decreases as the content of Cr^3+^ increases (Appendix A). 

In addition, the intense band located at around 46,729 cm^−1^ (214 nm) can be assigned to host absorption, and it moved to 44,643 cm^−1^ (224 nm) for the Mn-based sample. What is more, the bad is much broader because of overlapping with the Mn-O charge transfer band (CTB) (Appendix A). 

The crystal field Dq, Racah, B, and C parameters were calculated for Cr^3+^-doped samples (see Table 1) by using the same methodology as presented in reference [10]. Crystal field strength (CFS) Dq/B parameter is in the range of 2.29–2.39 for GAMn and 2.23–2.41 for GAMg samples. These results mean that Cr^3+^ ions are located in the intermediate ligand field, and energy separation between ^2^E and ^4^T_2g_ excited levels is not significant. The Dq/B parameter is slightly higher, around 2.41–2.43 for CAZn analogs. The calculated values of Dq/B are similar to those reported recently for DMANaCr (2.29) [15]. However, for some EA and DMA analogs (EANaCr 2.18 [7], EANaAlCr 2.21 [7], DMAKCr 2.21 [37], EAKCr 2.21 [37]) reported Dq/B values are much lower than for the investigated perovskites. On the other hand, the formate perovskites with AM^+^ cation comprising Cr^3+^ ions exhibit a strong crystal field (AMNaCr 2.743 [38], AMNaAlCr 2.55 [38]).

The emission spectra of investigated hybrid organic–inorganic formates [GA]M_1−x_Cr_x_(HCOO)_3_ (M = Mg^2+^, Mn^2+^, Zn^2+^, and x = 0.01, 0.03, 0.05) recorded at 80 K consists of the intense and narrow emission lines of Cr^3+^ ions located at 686 nm and 698 nm attributed to the spin-forbidden ^2^E → ^4^A_2g_ transitions (Figure 5). The broad emission band, which spans from 700 nm to 1000 nm, assigned to the spin-allowed transition from the ^4^T_2g_ excited level to the ^4^A_2g_ ground state is also observed [11,13,16,39]. As can be seen in Figure 5b,d and Appendix A, the emission intensity of GAMg and GAMn samples increased with the concentration of dopant ions, while the intensity of 1% Cr^3+^ and 5% Cr^3+^ in the GAZn analog are comparable. The samples with 3% of Cr^3+^ are out of the trend. However, the nature of this behavior is unspecified. The collation of the representative samples [GA]M_1−x_Cr_x_(HCOO)_3_ (M = Mg^2+^, Mn^2+^, Zn^2+^, and x = 0.05) showed that the most intense luminescence exhibits a sample comprising Mg^2+^ ions. The emission intensity of Mn^2+^ and Zn^2+^ samples is significantly less. The substitution of different metal M^2+^ ions in the crystal structure of guanidine formate have an impact on the intensity relationships between spin-forbidden and spin-allowed transition of Cr^3+^ ions. Only for the GAMg compound ^2^E → ^4^A_2g_ is emission more intense than spin-allowed transition; for GAMn and GAZn analogs, ^4^T_2g_ → ^4^A_2g_ transition dominates. It is worth noting that no emission of Mn^2+^ ions was detected, probably due to energy reabsorption by chromium ions.

The emission spectra in the function of temperature were recorded within the range of 80–300 K with 10 K steps. As can be seen in Figure 6 and Appendix A, the main component of the photoluminescence spectra belongs to the spin-allowed transitions of Cr^3+^ ions. Only for the GAMg sample containing 1% dopant, the ^2^E emission is much more intense than the band located at 795 nm. Generally, ^2^E → ^4^A_2g_ emission quenches significantly with increasing temperature, while the ^4^T_2g_ → ^4^A_2g_ emission of Cr^3+^ is more stable. It is due to the thermally stimulated energy transfer from ^2^E to ^4^T_2g_ energy level. Obtained results confirmed the occurrence of the intermediate ligand field in the nearest environment of Cr^3+^ ions. The mechanism of Cr^3+^ luminescence quenching is a well-known phenomenon in the literature and assumes crossing the ^4^T_2g_ excited state parabola with the ^4^A_2g_ one [11,13,16,39].

The significant dependence of photoluminescence intensity on temperature may be an interesting behavior that can be exploited for non-contact temperature readout based on luminescence. Figure 7 demonstrates a schematic representation of the approach to temperature determination. In this model, the Fluorescence Intensity Ratio (FIR) parameter can be defined as a ratio of the ^2^E → ^4^A_2g_ (spectral range 670–710 nm marked as I_1_) to the ^4^T_2g_ → ^4^A_2g_ (spectral range 750–1050 nm represented as I_2_) transition of Cr^3+^ ions, respectively. 

The proposed model was tested on the investigated [GA]M_1−x_Cr_x_(HCOO)_3_ (M = Mg^2+^, Mn^2+^, Zn^2+^, and x = 0.01, 0.03, 0.05) hybrid organic–inorganic perovskites. It is clear that the increase in temperature causes decreasing in FIR (Figure 8), and the highest value of FIR was obtained for the GAMg: 1% Cr^3+^ sample. To further comparison of the observed changes in thermometric parameters and to compare their features, the absolute (S_a_) and relative (S_r_) sensitivities were calculated as follows:(1)$$S_{a}=\left|\frac{dFIR}{dT}\right|,$$
and
(2)$$S_{r}=\frac{1}{FIR}\left|\frac{dFIR}{dT}\right|,$$
where *dFIR* represents the change of fluorescence intensity ratio at temperature change Δ*T*. The collation of S_a_ and S_r_ changes of the investigated hybrid organic–inorganic perovskites are presented in Figure 9 and Appendix A. Generally, the S_a_ and S_r_ values are the highest at the 80–120 K range and decrease with increasing temperature. However, the sensitivity changes with sample composition and concentration of Cr^3+^ ions. For GAMg: Cr^3+^ compounds, the significant relative sensitivity exhibits sample with the lowest concentration of dopant ions and equals 2.08%K^−1^ at 90 K. With increasing Cr^3+^ ions concentration, the *S_r_* decreased to around 1%K^−1^. Substitution of Mg^2+^ by Mn^2+^ caused a decrease of sensitivity to 1.20%K^−1^, but the optimal Cr^3+^ ions concentration was determined to be 3%. Similar trends are observed for GAZn: for Cr^3+^ analogs, however, the changes of S_r_ with chromium ions concentration are negligible, and the highest S_r_ is 1.08%K^−1^ at 90 K for GAZn: 1% Cr^3+^. Additionally, the repeatability of the thermal sensing performance of representative samples was verified by the circling heat/cool process (Appendix A). It can be seen that only a small variation from the initial value was observed, and the temperature parameter ∆ is reversed and repeated overheating/cooling cycles.

In fact, only one optical thermometer based on hybrid organic–inorganic formate perovskites [EA]_2_NaCr_0.21_Al_0.79_(HCOO)_6_ with a sensitivity S_r_ of 2.84%K^−1^ at 160 K is known [10]. Obtained values of relative sensitivities were compared with the S_r_ values of other inorganic and hybrid organic–inorganic luminescent thermometers (Table 2). The results show that investigated [GA]M_1−x_Cr_x_(HCOO)_3_ (M = Mg^2+^, Mn^2+^, Zn^2+^, and x = 0.01, 0.03, 0.05) has the potential to be applied as a low-temperature luminescent thermometer. 

## 4. Conclusions

Three series of samples [C(NH_2_)_3_]M(HCOO)_3_ (Mg^2+^, Mn^2+^, Zn^2+^) doped with 1%, 3%, and 5% of Cr^3+^ ions were synthesized using the low-diffusion synthesis method. Their structural, phonon, and luminescent properties were investigated in detail. It was shown that the incorporation of Cr^3+^ ions into the crystal structure of investigated hybrid organic–inorganic perovskites does not affect the phase purity of the samples. Based on diffuse reflectance spectra, crystal field strength (Dq/B) and Racah parameters were determined. It was found that Cr^3+^ ions are located in the intermediate crystal field or close to it with a Dq/B range of 2.29–2.41. The investigation of sample composition showed that the highest emission intensity exhibits GAMg: 5% Cr^3+^ sample, while the lowest one GAZn: 5% Cr^3+^. The presence of both the spin-forbidden and spin-allowed transitions of Cr^3+^ ions at a broad temperature range enables the characterization of these materials as luminescence thermometers. It turned out that the relative sensitivity of S_r_ depends on the sample composition and concentration of Cr^3+^ ions. The highest relative sensitivity S_r_ = 2.08%K^−1^ at 90 K has [GA]Mg(HCOO)_3_: 1% Cr^3+^. Replacement of Mg^2+^ by Mn^2+^ or Zn^2+^ reduced the sensitivity to 1.20%K^−1^ at 100 K and 1.08%K^−1^ at 90 K for [GA]Mn(HCOO)_3_: 3% Cr^3+^ and [GA]Zn(HCOO)_3_: 1% Cr^3+^, respectively.

## Figures and Tables

**Figure 1 sensors-23-06259-f001:**
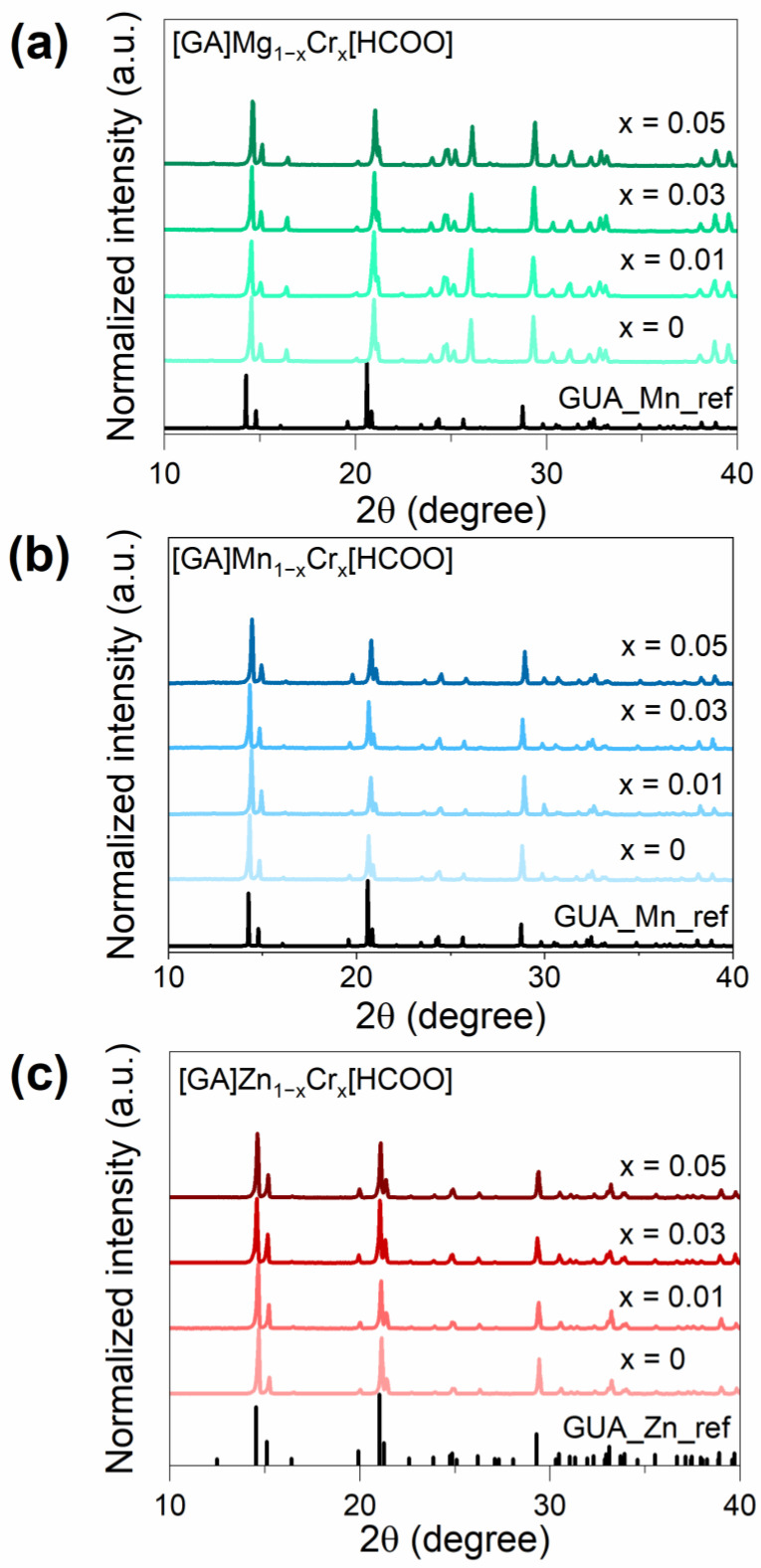
XRD patterns for a series of [GA]Mg_1−x_Cr_x_(HCOO)_3_ (x = 0, 0.01, 0.03, 0.05) (**a**), [GA]Mn_1−x_Cr_x_(HCOO)_3_ (x = 0, 0.01, 0.03, 0.05 (**b**), and [GA]Zn_1−x_Cr_x_(HCOO)_3_ (x = 0, 0.01, 0.03, 0.05 (**c**).

**Figure 2 sensors-23-06259-f002:**
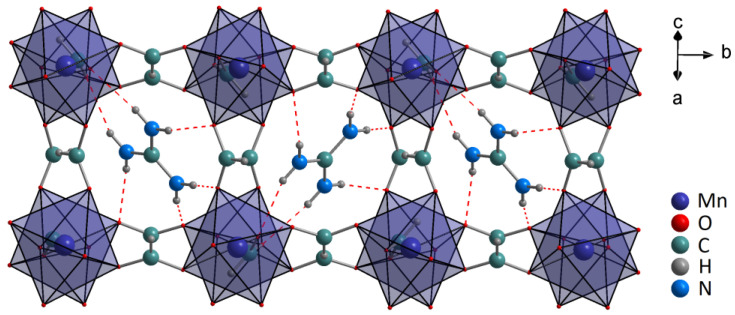
The crystal structure of [GA]Mn(HCOO)_3_ based on data presented in [31]. The dashed lines present HBs between GA^+^ cations and the manganese-formate framework.

**Figure 3 sensors-23-06259-f003:**
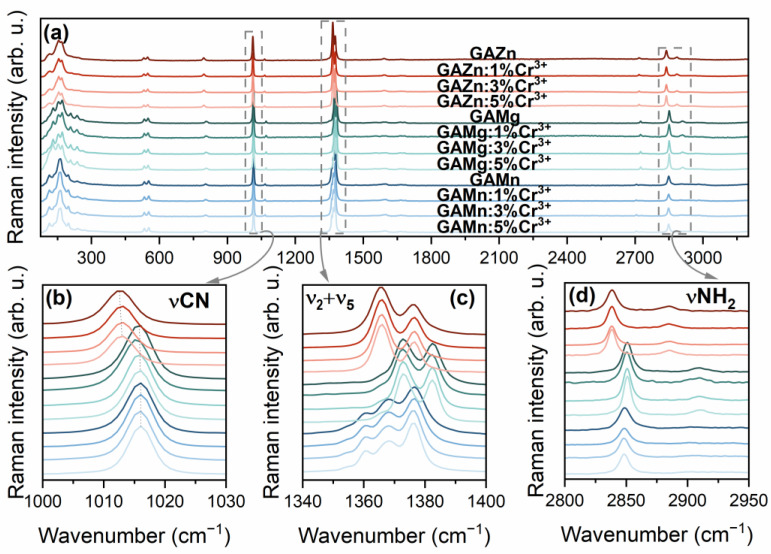
The Raman spectra of the [GA]M_1−x_Cr_x_(HCOO)_3_ series, where M = Mg^2+^, Mn^2+^, Zn^2+^, and x = 0, 0.01, 0.03, and 0.05 (**a**) and the enlarged areas with bands corresponding to stretching C–N (**b**), symmetric C–O stretching and C–H in-plane bending (ν_2_ + ν_5_) (**c**), and stretching N–H modes (**d**).

**Figure 4 sensors-23-06259-f004:**
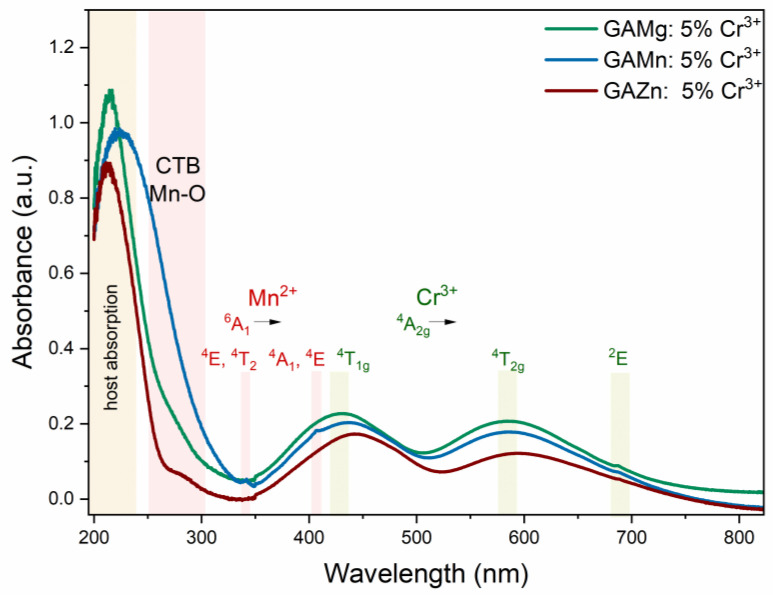
Diffuse reflectance spectra of representative samples [GA]M_1−x_Cr_x_(HCOO)_3_ (M = Mg^2+^, Zn^2+^, Mn^2+^ and x = 0.05) measured at 300 K.

**Figure 5 sensors-23-06259-f005:**
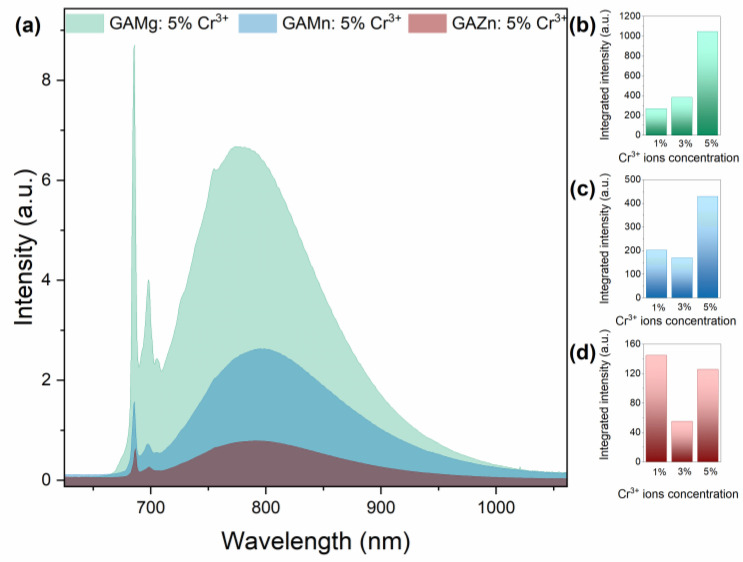
Emission spectra of [GA]M_1−x_Cr_x_(HCOO)_3_ (M = Mg^2+^, Mn^2+^, Zn^2+^, and x = 0.05) at 80 K (**a**) and influence of Cr^3+^ ions concentration of emission intensity (**b**–**d**) of the investigated samples.

**Figure 6 sensors-23-06259-f006:**
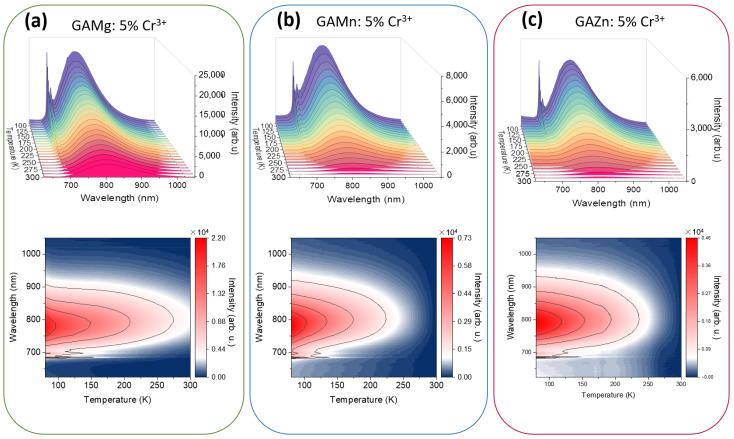
Temperature-dependent emission spectra and thermal evolution of emission intensity of [GA]Mg(HCOO)_3_: 5% Cr^3+^ (**a**), [GA]Mn(HCOO)_3_: 5% Cr^3+^ (**b**), and [GA]Zn(HCOO)_3_: 5% Cr^3+^ (**c**) representative samples, respectively.

**Figure 7 sensors-23-06259-f007:**
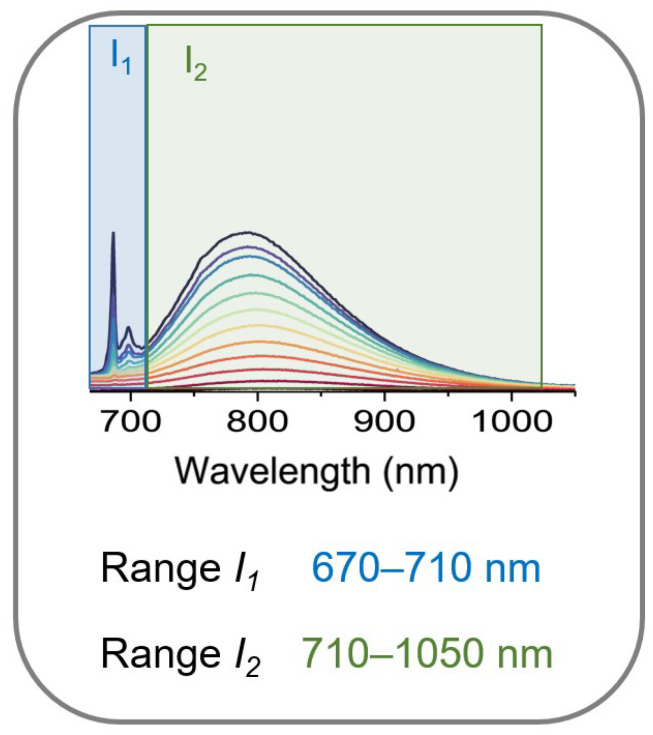
Graphical representation of way for the temperature detection in hybrid organic–inorganic formate perovskites [GA]M_1−x_Cr_x_(HCOO)_3_ (M = Mg^2+^, Mn^2+^, Zn^2+^, and x = 0.01, 0.03, 0.05).

**Figure 8 sensors-23-06259-f008:**
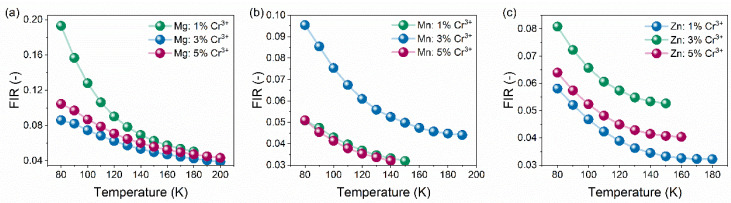
Influence of Cr^3+^ ions concentration on Fluorescence Intensity Ratio (FIR) (**a**–**c**) of [GA]M_1−x_Cr_x_(HCOO)_3_ (M = Mg^2+^, Mn^2+^, Zn^2+^, and x = 0.01, 0.03, 0.05) hybrid perovskites.

**Figure 9 sensors-23-06259-f009:**
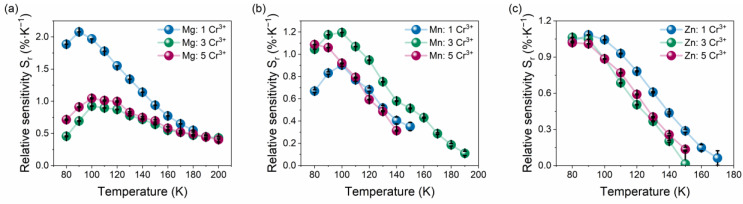
Influence of Cr^3+^ ions concentration on relative sensitivity S_r_ (**a**–**c**) of [GA]M_1−x_Cr_x_(HCOO)_3_ (M = Mg^2+^, Mn^2+^, Zn^2+^, and x = 0.01, 0.03, 0.05) hybrid perovskites.

**Table 1 sensors-23-06259-t001:** The collation of crystal field parameters and energies of electron transitions of the investigated series of [GA]M_1−x_Cr_x_(HCOO)_3_ (M = Mg^2+^, Mn^2+^, Zn^2+^, and x = 0.01, 0.03, 0.05).

Parameters	GAMn:			GAMg:			GAZn:		
	1%Cr^3+^	3%Cr^3+^	5%Cr^3+^	1%Cr^3+^	3%Cr^3+^	5%Cr^3+^	1%Cr^3+^	3%Cr^3+^	5%Cr^3+^
^4^A_2g_–^2^E (cm^−1^)	14,535	14,536	14,537	14,552	14,552	14,547	14,540	14,539	14,540
^4^A_2g_–^4^T_2g_ (cm^−1^)	15,545	15,959	15,735	15,828	15,917	16,259	15,640	15,544	15,500
^4^A_2g_–^4^T_1g_ (cm^−1^)	21,972	22,156	22,439	22,703	22,682	22,952	22,062	21,901	21,869
Dq (cm^−1^)	1555	1555	1574	1583	1592	1626	1564	1554	1550
B (cm^−1^)	650	675	686	709	692	676	648	641	643
Dq/B	2.39	2.30	2.29	2.23	2.30	2.41	2.41	2.43	2.41
C (cm^−1^)	3242	3190	3166	3122	3157	3184	3247	3264	3259
C/B	4.13	4.25	4.62	4.40	4.57	4.71	5.01	5.09	5.07

**Table 2 sensors-23-06259-t002:** Collation of exemplary luminescent thermometers with their highest relative sensitivity (S_r_) at working temperature (T) ^1^.

**Compound**	**S_r_ (%K^−1^)**	**T (K)**	**Reference**
[GA]Mg(HCOO)_3_: 1% Cr^3+^	2.08	90	This work
[GA]Zn(HCOO)_3_: 1% Cr^3+^	1.08	90	This work
[GA]Mn(HCOO)_3_: 3% Cr^3+^	1.20	100	This work
[EA]_2_NaCr_0.21_Al_0.79_(HCOO)_6_	2.84	160	[10]
(Me_2_NH_2_)_3_[Eu_3_(FDC)_4_(NO_3_)_4_]·4H_2_O	2.7	170	[40]
Sr(HCOO)_2_:Eu^2+^/Eu^3+^	3.8	293	[41]
Ln-cpda (Ln = Eu, Tb)	16	300	[42]
TbMOF@3%Eu-tfac	2.59	225	[43]
[Eu_2_(qptca)(NO_3_)_2_(DMF)_4_](CH_3_CH_2_OH)_3_perylene	1.28	293	[44]
Bi_2_Ga_4_O_9_:Cr^3+^	0.7	290	[45]
Bi_2_Al_4_O_9_:Cr^3+^	1.24	290	[46]
Sr_2_MgAl_22_O_36_:Cr^3+^	1.7	310	[47]
ZnGa_2_O_4_:Cr^3+^	2.8	310	[48]
SrAl_12_O_19_:Mn^4+^	0.27	393	[49]
LaPO_4_:Nd^3+^	7.19	303	[50]
MgTiO_3_:Mn^4+^	1.2	93	[51]
La_2_MgTiO_6_: Cr^3+^, V^4+^	1.96	165	[11]

^1^ GA—guanidine, EA—ethylammonium, H_2_FDC—9-fluorenone-2,7-dicarboxylic acid, H_3_cpda—5-(4-carboxyphenyl)-2,6-pyridinedicarboxylic acid, TbMOF—[Tb_2_(bpydc)_3_(H_2_O)_3_]·*n*DMF, H_2_bpydc—2,2-bipyridine-5,5′-dicarboxylic acid, tfac—trifluoroacetylacetonate, H_4_qptca—1,1′:4′,1′′:4′′,1′′′-quaterphenyl-3,3′′′,5,5′′′-tetracarboxylic acid, DMF—dimethylformamide.

## Data Availability

Experimental data: The Raman and diffuse reflectance spectra, temperature-dependent luminescence and emission maps, thermometric parameters, powder XRD data, and low-temperature emission spectra are available at 10.5281/zenodo.7970355.

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
