# Peer review of "Structure, Luminescence and Temperature Detection Capability of [C(NH2)3]M(HCOO)3 (M = Mg2+, Mn2+, Zn2+) Hybrid Organic–Inorganic Formate Perovskites Containing Cr3+ Ions"

_sensors, 2023, doi:10.3390/s23146259_

Round 1
Reviewer 1 Report
Luminescent thermometry is relatively new and very perspective field of application of hybrid perovskites. Their application enables to go away from rare-earth containing materials that is important in industrial scale because of the high price of RE compounds. The authors have synthesized and proved the opportunity to use as non-contact luminescent thermometers of some Cr(3+)-doped divalent metal formates with guanidinium counter-cation. The relative sensitivity of 2.08% K-1 has been achieved that is comparable with some RE-containing materials.
Despite the fact that the results obtained are really of interest and really can be used in practice, some drawbacks of the paper should be mentioned.
1. The choice of Mg(2+), Mn(2+) and Zn(2+) as divalent cations and of guanidinium cation as counter-ion should have some background. Please, describe which properties of these ions (i.e., the possible participation of guanidinium in hydrogen bond formation) have determined this choice.
2. Why hybrid perovskites based on formate anion exhibit unique properties?
3. From XRD patterns (fig. 1) the one for Zn-compound seems to be a little bit different. Also I recommend to include the figure with crystal structure to see both crystal packing important for MOF and also hydrogen bonds network.
4. Raman spectra for Mn-compound (Fig. 2b, c) seems to be different from others in number of components. That can not arise from the difference in cell parameters. By the way, I recommend the authors to add the table with lattice parameters into the text.
5. Sometimes the authors include x = 0 into legend to figures of into the description of the luminescent properties though compounds without Cr(3+) dopant do not exhibit luminescent properties (see, for example, legend to Fig. 7, description of emission spectra on lines 197-198, 275-276 etc.).
6. The authors have mentioned difference in hydrogen bonding in compounds studied. I would like to see some discussion of its effect on luminescent properties.
7. The legend to table 2 should be changed: as to I understand, the highest values of relative sensitivity and corresponding temperatures are mentioned.
Reviewer 2 Report
The authors prepared a new perovskite-like organic-inorganic materials (combination of guadinium, divalent metal ion and formate anion) and doped it with various amount of Cr3+. The materials were characterized by means of X-ray diffraction, vibration spectroscopy and UV-Vis spectroscopy and their luminescence properties were studies. The materials show temperature-dependent luminescence, and can be potentially used as optical thermosensors. The work was done carefully, the results are presented clearly and the manuscript is well organized and clearly written (but with some inconsistencies). As it brings new and interesting results, I in general recommend its publication. However, there are some (mostly minor) objections, which should be clarified before the final publication. They are:
Formal remarks:
line 28: NH4(1+) is not organic cation, please, re-formulate
line 126: cation M was substituted by Cr3+, not in opposite
Factual remarks:
line 171: rather “very weak”
line 174: in the case of Mn-phase, the shift is to lower wavenumber, in the case of Mg-phase, it stays at the same position, for Zn, it is not seen... please, describe the spectra S1-S3 correctly or add expansions of the spectra – it cannot be easily distinguished by eye in the present form
line 176: the “very sharp and intense” peak at 29240 cm(-1) cannot be seen in the spectra S1-S3 as they ends at 28500 cm(-1)... please, enlarge the scale... In the Fig 3, there is only a very weak band...
What is “CTB” in Fig. 3? There is no resolved band...
line 185: Dq/B of 2.3 for GAMn and GAMg is not true in general, GAMn with 1%Cr has 2.39, GAMg with 5%Cr 2.41, fully comparable with GAZn... formulate more carefully...
line 208: how was emission intensity quantified? (in some “absolute” units?) ok, it seems, that intensity for GAMg is in general the highest when compared to others, but also within the group, the intensities vary non trivially with content of Cr: for GAMg, there is expected trend, but for GAMn, the lowest intensity was observed for 3%Cr, and for Zn, the trend was even fully opposite... so, if the trends are not clear within group of materials of one “main” metal ion, are you sure about general statement comparing the intensities between the groups based on different metals (Mn, Mg, Zn)?
Figure S8 should be moved to the main text, as it clearly shows the dependence of shape of fluorescence spectra on temperature... Figure 7 shows just “derivation”, which is important for determining optimal temperature range for sensor use, but is not such illustrative...
Please, check the value od derivation at 80 K... How was it calculated? (I.e., how was determined dT, if this is the starting point on the temperature scale?)
Main: how many experiments was performed with individual phases? I.e., what is reproducibility? It should be stated.
Reviewer 3 Report
This paper deals with a series of inorganic formate perovskites that are doped with varying amounts of Cr3+ ion, and it reports results on the observed long-wavelength luminescence of these systems. It is found that the sample composition and incorporation of Cr3+ provide new materials, where affects the temperature dependence significantly influences the luminescence intensity. Hence, these materials can be considered "luminescence thermometers".
The work appears to be well done, both experimentally and in its interpretation. Please could the authors clarify if the reported emission spectra, detected with the Hamamatsu PMA-12 photonic multichannel analyzer and BT-CCD sensor, are corrected for wavelength dependence? Also, the penultimate sentence in the abstract, beginning "The relative sensitivity of Sr..." seems confusing, as the relative sensitivity is actually defined as Sr.
Round 2
Reviewer 1 Report
Now I am satisfied by the paper, the authors have made the corrections necessary.